# Targeted Protein Profiling of In Vivo NIPP-Treated Tissues Using DigiWest Technology

**Felix Ruoff** [1], **Melanie Henes** [2], **Markus Templin** [1], **Markus Enderle** [3], **Hans Bösmüller** [4], **Diethelm Wallwiener** [2], **Sara Y. Brucker** [2], **Katja Schenke-Layland** [1,5,6,7] **and Martin Weiss** [1,2,*]

1   NMI Natural and Medical Sciences Institute, University of Tübingen, 72770 Reutlingen, Germany; Felix.Ruoff@nmi.de (F.R.); markus.templin@nmi.de (M.T.); kschenkelayland@me.com (K.S.-L.)
2   Department of Women's Health, University of Tübingen, 72076 Tübingen, Germany; melanie.henes@med.uni-tuebingen.de (M.H.); Diethelm.Wallwiener@med.uni-tuebingen.de (D.W.); sara.brucker@med.uni-tuebingen.de (S.Y.B.)
3   Erbe Elektromedizin GmbH, 72072 Tübingen, Germany; Markus.Enderle@erbe-med.com
4   Department of Pathology and Neuropathology, Eberhard Karls University, 72076 Tübingen, Germany; hans.boesmueller@med.uni-tuebinge.de
5   Department of Biomedical Engineering, Eberhard Karls University, 72076 Tübingen, Germany
6   Cluster of Excellence iFIT (EXC 2180) "Image-Guided and Functionally Instructed Tumor Therapies", Eberhard Karls University, 72074 Tübingen, Germany
7   Department of Medicine/Cardiology, University of California, Los Angeles, CA 90095, USA
*   Correspondence: martin.weiss@med.uni-tuebingen.de; Tel.: +49-7071-29-82211

**Abstract:** Non-invasive physical plasma (NIPP) is a novel therapeutic tool, currently being evaluated for the treatment of cancer and precancerous lesions in gynecology and other disciplines. Additionally, patients with cervical intraepithelial neoplasia (CIN) may benefit from NIPP treatment due to its non-invasive, side-effect-free, and tissue-sparing character. However, the molecular impact of in vivo NIPP treatment needs to be further investigated. For this purpose, usually only very small tissue biopsies are available after NIPP treatment. Here, we adapted DigiWest technology, a high-throughput bead-based Western blot, for the analysis of formalin-fixed paraffin-embedded (FFPE) cervical punch biopsies with a minimal sample amount. We investigated the molecular effects of NIPP treatment directly after (0 h) and 24 h after in vivo application. Results were compared to in vitro NIPP-treated human malignant cervical cells. NIPP effects were primarily based on an inhibitory impact on the cell cycle and cell growth factors. DigiWest technology was suitable for detailed protein profiling of small, primary FFPE biopsies.

**Keywords:** FFPE protein extraction; non-invasive physical plasma; DigiWest; CIN; in vivo treatment



## 1. Introduction

Physical plasma is defined as a highly energized gas, forming reactive oxygen and nitrogen species (ROS and RNS) by interacting with the atmosphere, fluids, and organic surfaces. Consequently, ROS and RNS cause distinct cellular responses, including anti-proliferative and apoptotic cell mechanisms [1,2]. This enables the induction of protherapeutic biomedical effects regarding precancerous and cancerous tissue. In recent studies, non-invasive physical plasma (NIPP) treatment offered promising anti-neoplastic effects on a wide range of tumors in the field of gynecology and other medical subspecialties [3–9].

Cervical intraepithelial neoplasia (CIN) are very frequent precancerous lesions in young women, which may lead to cervical cancer. Thus, cervical cancer is still the fourth most common cancer for women worldwide, with about 270,000 cancer-related deaths per year [10,11]. Despite the fact that only few CIN lesions become invasive in the end, current guidelines recommend local excision procedures, which are associated with invasiveness, the need for local or general anesthesia, and serious short- and long-term side effects and risks, especially during pregnancy [12,13]. Therefore, overtreatment is a problem

for affected women and health economy. Recently, we deeply characterized NIPP as an innovative, non-invasive treatment procedure for CIN treatment [14].

To date, most of the knowledge about NIPP-related effects on human cells originates from in vitro experiments. To improve our understanding about the mode of action and about the conceivable medical applications of this innovative treatment approach it is important to investigate the molecular NIPP effects within a patient and to gain the maximum amount of possible information from in vivo NIPP-treated and formalin-fixed paraffin-embedded (FFPE) small tissue samples. Since the 1980s and the 1990s, it has been possible to extract and subsequently analyze DNA and RNA from FFPE tissue. This technique is even used in clinical routine nowadays [15]. Yet, to further enhance functional precision medicine it is essential to move beyond pure genetic and transcriptional analysis. Post-transcriptional regulatory mechanisms can have a tremendous impact on the molecular function of cells, and malfunctions induced through changes in protein level can be missed by pure genomic approaches [16]. To make it worse, cervical punch biopsies are characterized by a very small sample size (Figure 1).

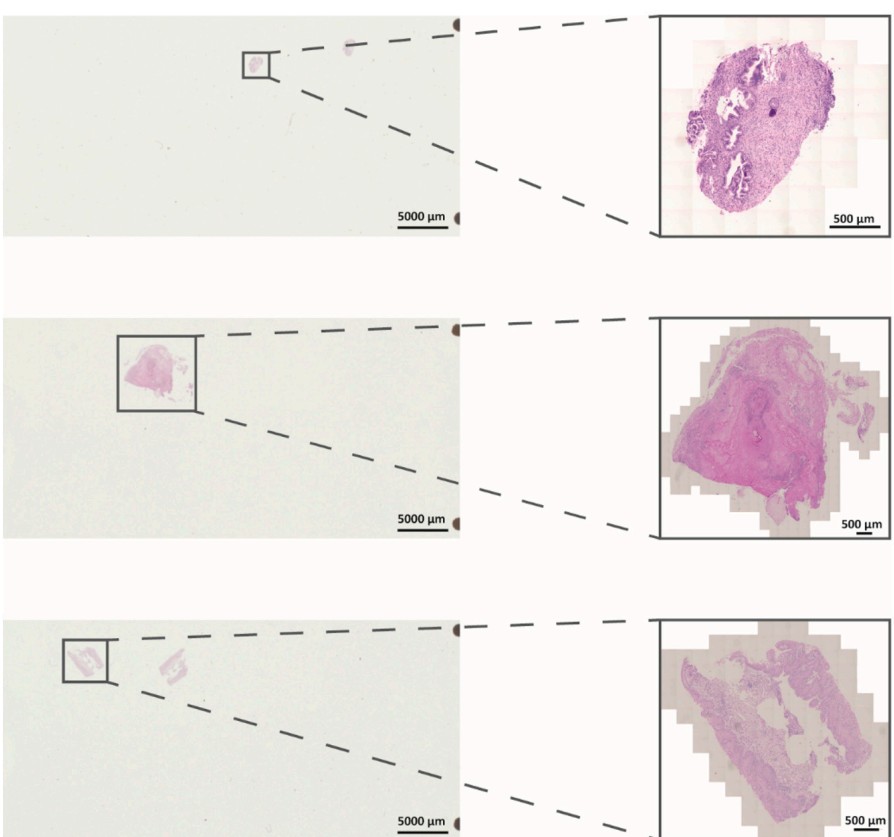

**Figure 1.** Representative light microscopic H&E staining of FFPE samples on slides. Shown are overview pictures of slides and magnified tissue areas (black boxes). Pictures were taken utilizing an Axio Scan Z.1 (Zeiss).

Hence, we established a workflow for targeted protein analysis from FFPE tissue samples from cervical punch biopsies by adapting a commercially available lysis protocol and utilizing DigiWest technology to obtain valuable molecular insights regarding in vivo NIPP treatment of CIN [17].

## 2. Materials and Methods

### 2.1. In Vitro and In Vivo NIPP Treatment

For NIPP treatment the electrosurgical device VIO® 3, APC 3 (Erbe Elektromedizin, Tübingen, Germany) was used (argon gas flow: 1.6 L/min; preciseAPC, effect 1). Cells were

dynamically treated in a suspension on a 6-well cell culture plate in 700 μL of DMEM at a distance of 7 mm. The NIPP treatment of single cells was performed in a suspension for the following reasons: (i) to avoid mechanical detachment and associated cell damage, as well as drying effects due to the NIPP gas flow, and (ii) to enable NIPP treatment on a sterile, grounded metal mold with an identical electrical current and resistance. It was recently shown that the activation of media by plasma (plasma-activated media; PAM) reveals very similar anti-proliferative cell effects compared to direct plasma treatment [3]. Moreover, the unphysiological suspension state was limited to the treatment period before enabling the immediate reattachment of the cells. According to NIPP treatment, the controls were treated with argon gas alone (flow: 1.6 L/min) to exclude any alterations in cells and tissues due to the treatment procedure.

NIPP treatment of patients. Before NIPP treatment a clinical examination by colposcopy, 4% acetic acid, and Lugol's iodine staining was performed followed by NIPP treatment under a colposcopic view of visualized HSIL. The lesions were treated with NIPP for 30 s/cm2 using a VIO3/APC3 and 3.2 mm APC probes (settings: preciseAPC, effect 1; ERBE Elektromedizin). The treatment was carried out on an outpatient basis and without local or general anesthesia on a conventional gynecological examination chair. A commercially available, reusable silicone electrode mat was placed under the patient as a negative electrode. The NIPP probe was passed over the tissue in defined "brush strokes" in order to avoid localized heating of the tissue. Treatment was performed within an ongoing prospective, single-armed phase IIb clinical trial (NCT03218436) at the Department for Women's Health, Tübingen, Germany. NIPP treatment and tissue analysis was approved by the Ethical Committee of the Medical Faculty of the Eberhardt Karls University of Tübingen (237-2017BO1).

### 2.2. Propagation of Cells

Cervical squamous cell carcinoma-derived (CSCC) cells were purchased from ATCC (ATCC® TCP-1022™, American Type Culture Collection). In detail, these were CaSki (ATCC CRL-1550), DoTc2-4510 (ATCC CRL-7920), and SiHa (ATCC HTB-35). CaSki and SiHa cells are positive for human papillomavirus (HPV) and are derived from squamous cell carcinomas of the cervix uteri, whereas DoTc2 4510 cells are derived from adenocarcinomas. CSCC cells were cultured in Dulbecco's Modified Eagle's Medium (DMEM F12, Fischer Scientific, Hampton, NH, USA), supplemented with 10% fetal bovine serum (Life Technologies, Carlsbad, CA, USA), 1 mM of sodium pyruvate (Life Technologies), and 1% penicillin/streptomycin (Invitrogen, Carlsbad, CA, USA) at 37 °C and 5% $CO_2$ in a humidified atmosphere. Every 2–3 days, a medium exchange was performed, and cells were passaged after reaching 70%–80% confluence. The adherent cells were detached by trypsin/EDTA (0.05%, 10 mM at 37 °C; Life Technologies) treatment.

### 2.3. Protein Extraction from FFPE Tissue

Macrodissection of FFPE tissue (10–400 mm$^2$) mounted on 4–6 slides was performed utilizing a Qproteom FFPE Tissue kit (Quiagen, Hilden, Germany). Tissue picks (Covaris, Woburn, MA, USA) were moistened with 2 μL of ExB+ (Quiagen, Hilden, Germany) and used to scrape off desired tissue areas. An H&E-stained master slide, marked by a pathologist, was used as a template. Samples were collected in 1.5 mL LoBind reaction tubes (Eppendorf, Hamburg, Germany). For protein extraction, the heptane-based protocol was used. The volume of ExB+ was adjusted to 20 μL; all other steps were performed according to the manufacturer's recommendations. The volume of protein lysates was reduced using a vacuum concentrator for 1.5 h. The resulting protein lysates were diluted in loading buffer containing 212 mM of Tris, 282 mM of Tris base, 1.01 mM of EDTA, and 50 mM of DTT (Invitrogen, Carlsbad, CA, USA), supplemented with 10% glycerol, 0.22 mM of Coomassie brilliant blue, and 0.175 mM of phenol red (Figure S1a).

## 2.4. Lysis of Cell Culture Pellets

Dry cell pellets were lysed by adding 30 μL of a lysis buffer, containing 4% SDS, 50 mM of DTT, cOmplete protease inhibitor, and PhosSTOP phosphatase inhibitor (Roche, Basel, Switzerland) on ice, and by subsequently being incubated for 10 min at 95 °C in a block heater. The samples were brought to room temperature, and the whole volume was transferred to a QuiaShredder spin column (Quiagen, Hilden, Germany) and then centrifuged at $16,000\times g$ for 5 min to remove DNA. Samples were transferred to and stored in 1.5 mL LoBind reaction tubes (Eppendorf, Hamburg, Germany).

## 2.5. Multiplex Protein Profiling via DigiWest

DigiWest was performed as described previously [18]. Briefly, the NuPAGE system (Life Technologies, Carlsbad, CA, USA) with a 4–12% Bis-Tris gel was used for gel electrophoresis and Western blotting onto PVDF membranes. After washing with PBST, proteins were biotinylated by adding 50 μM of NHS-PEG12-Biotin in PBST for 1 h to the membrane. After washing in PBST, membranes were dried overnight. Each Western blot lane was cut into 96 strips of 0.5 mm each. Strips of one Western blot lane were sorted into a 96-well plate (Greiner Bio-One, Frickenhausen, Germany) according to their molecular weight. Protein elution was performed using 10 μL of elution buffer (8 M urea, 1% Triton-X100 in 100 mM of Tris-HCl with a pH of 9.5). Neutravidin-coated MagPlex beads (Luminex, Austin, TX, USA) of a distinct color ID were added to the proteins of a distinct molecular weight fraction, and coupling was performed overnight. Leftover binding sites were blocked by adding 500 μM of deactivated NHS-PEG12-Biotin for 1 h. To reconstruct the original Western blot lane, the beads were pooled, at which point the color IDs represent the molecular weight fraction of the proteins.

For antibody incubation, 5 μL aliquots of the DigiWest bead mixes were added to 50 μL of an assay buffer (blocking reagent for ELISA (Roche, Rotkreuz, Switzerland) supplemented with 0.2% milk powder, 0.05% Tween-20, and 0.02% sodium azide) or PVXC buffer (0.1% casein, 0.5% PVA, 0.8% PVP, and 0.05% Tween-20 in PBS) in a 96-well plate. The buffer was discarded and 30 μL of primary antibody diluted in assay buffer or PVXC buffer was added per well. Primary antibodies were incubated overnight at 15 °C on a shaker. Subsequently, primary antibodies were discarded and beads were washed twice with PBST. After washing, 30 μL of species-specific secondary antibody diluted in an assay buffer or PVXC buffer labeled with phycoerithrin was added and incubation took place for 1 h at 23 °C. Before the readout on a Luminex FlexMAP 3D, beads were washed twice with PBST. Protein bands were depicted as peaks by plotting the molecular weight to the corresponding median signal intensity. An Excel macro-based algorithm was used to identify peaks at the provided molecular weight of each antibody. After subtracting the local background integrals of the area of a peak was calculated. The resulting signals were normalized to the total amount of protein that was loaded onto the beads (Figure S1b).

## 2.6. Statistical Analysis

Statistical comparison was carried out with a Wilcoxon rank-sum test or a Kruskall–Wallis test, (GraphPad Prism version 6.0, GraphPad Software; MultiExperiment Viewer (MeV) version 4.0.9 [19]), as specified in the figure legends. The data are expressed as mean $\pm$ standard deviation. $p$ values of $<0.05$ were considered statistically significant.

## 3. Results

### 3.1. Establishment and Evaluation of DigiWest from FFPE Samples after In Vivo NIPP Treatment

Due to the very small sample size of cervical punch biopsies, multiplex protein profiling from FFPE tissue obtains valuable molecular insights into NIPP treatment. To analyze protein expression levels after in vivo NIPP treatment we established DigiWest technology to FFPE tissue slides before ($n = 5$ patients), directly after (0 h; $n = 3$ patients), and 24 h after in vivo NIPP treatment ($n = 4$ patients). The tissue specimens used were

mounted on slides for histopathological assessment, and the tissue area ranged from 10 mm² to 400 mm² (Figure 1).

Using DigiWest, 69 analytes covering apoptosis machinery, DNA damage response (DDR), and cell cycle control were analyzed. Resulting median fluorescence intensity (MFI) values were compared between samples before, 0 h after treatment, and a control, as well as 24 h after treatment and a control. Twenty-nine antibodies delivered a detectable signal in all samples. After median centration and log2 transformation, hierarchical clustering (complete linkage, Euclidean distance) revealed a similar protein expression pattern of all preNIPP samples (Figure 2). Yellow indicates a high signal level, whereas blue indicates a low signal level when compared to the median of all samples. Samples are clustered horizontally and analytes are clustered vertically. Most of the samples 0 h and 24 h postNIPP have a similar protein expression profile. This indicates a good sample quality and protein yield.

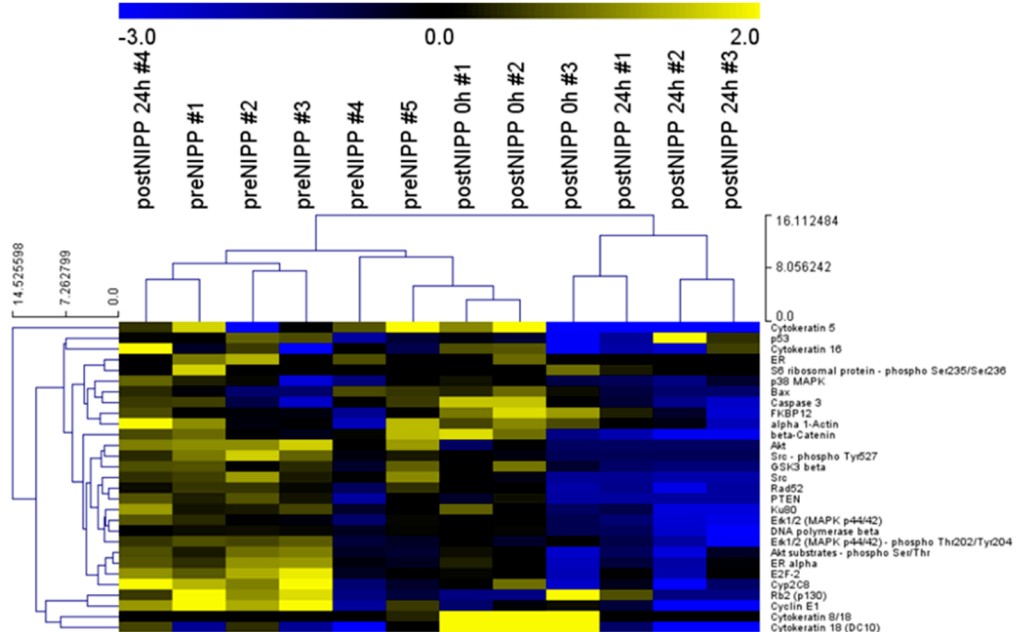

**Figure 2.** DigiWest protein profiles of a control (untreated) as well as 0 h and 24 h after treatment patient samples. Heat map of log2 transformed DigiWest data. Data were median-centered, and hierarchical clustering was performed using complete linkage and Euclidian distance, utilizing the MultiExperiment Viewer (MeV version 4.9.0, ref. [19]) software. Yellow indicates high signal level and blue indicates a low signal level (compared to the median).

### 3.2. Protein Profiles of Patients and Cell Culture following In Vivo and In Vitro NIPP Treatment

NIPP treatment induces various biological effects, including antineoplastic efficacy [8]. Therefore, NIPP is a promising tool for the treatment of precancer and cancer. Here, we examined the overall antineoplastic properties by performing both in vitro NIPP treatment of the human malignant cervical cell line, CSCC, as well as in vivo NIPP treatment of patients with histologically confirmed lesions of CIN.

First, we analyzed cell pellets from an NIPP-treated CSCC cell culture (*n* = 6) compared to argon-treated controls (*n* = 6) harvested after 24 h. We analyzed a total of 132 proteins, covering apoptosis, DDR, and cell cycle control, forty-four of which were matching analytes with the FFPE analysis (Figures 3a and 4a). Generally, the differences in signal intensity in the cell culture sample set was rather low, due to only one analyte showing a mean log2 foldchange greater than 1 or −1 after in vitro NIPP treatment (Figure 3b).

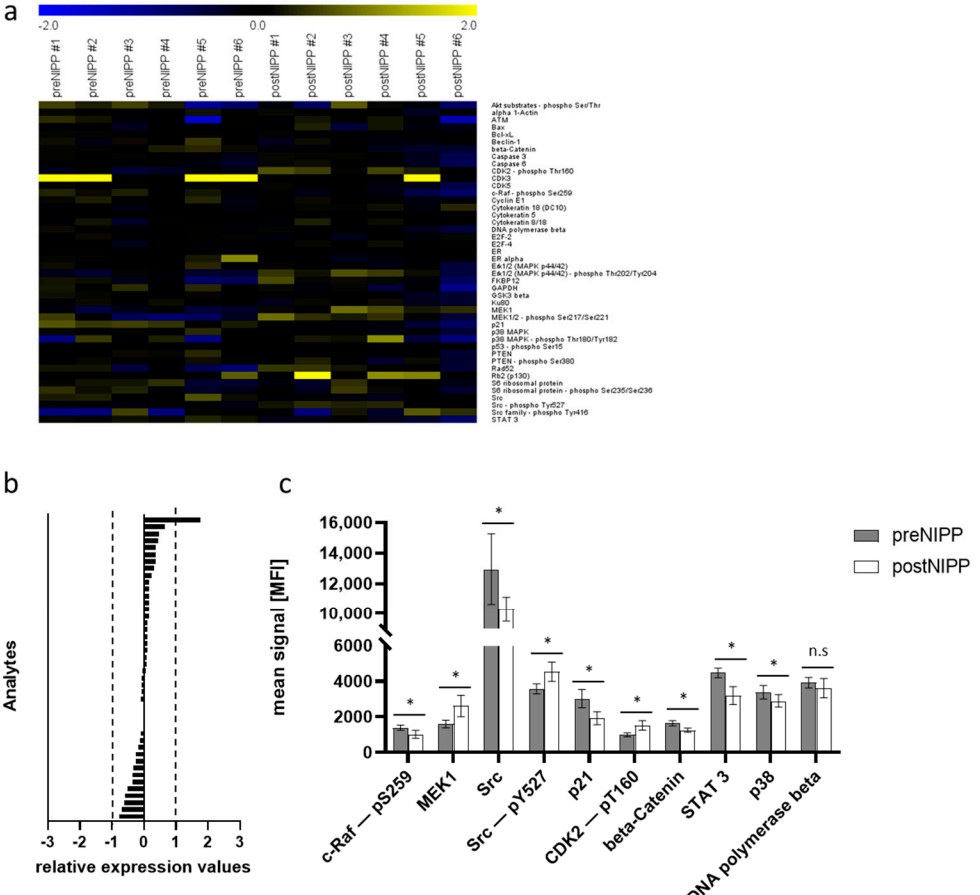

**Figure 3.** Protein expression after in vitro non-invasive physical plasma (NIPP) treatment of the human malignant cervical squamous cell carcinoma cell line (CSCC). (**a**) Heat map of log2-transformed DigiWest data. Yellow indicates a high signal level and blue indicates a low signal level (compared to the median). (**b**) Bar graphs of log2-transformed ratios calculated from the mean expression of NIPP treatment and controls, sorted from the highest positive change to the highest negative change. (**c**) Bar graphs of signals generated from cell culture samples. Shown are the means and standard derivation of NIPP treatment and controls. * indicates a significant difference in expression, n.s. indicates no significant difference in expression (Wilcoxon rank sum test, $p < 0.05$).

NIPP-treated CSCC cells showed significantly reduced expression levels of various pro-proliferative factors (Figure 3c). We found that the mitogen-activated protein kinase (MAPK) pathway was targeted by NIPP treatment. This was shown by a significant downregulation of p38 mitogen-activated protein kinases (p38), the RAF proto-oncogene serine/threonine-protein kinase (c-Raf) acting as a kinase cascade initiator [20], as well as the dual specificity mitogen-activated protein kinase kinase 1 (MEK1), being a dual threonine and tyrosine recognition kinase responsible for MAPK phosphorylation and activation [21]. These factors are critically involved in cell growth and apoptosis regulation and can act as oncogenes. Furthermore, the proto-oncogene tyrosine-protein kinase Src was significantly decreased, whereas it showed a slight but statistically significant increase in phosphorylation (pTyr527). As a central proto-oncogene, Src plays an important role in cell survival, proliferation, and invasion, and has been shown to interact with several signaling pathways, including MAPK/MEK1/RAF, Akt, and signal transducer and activator of transcription 3 (STAT3) [22,23].

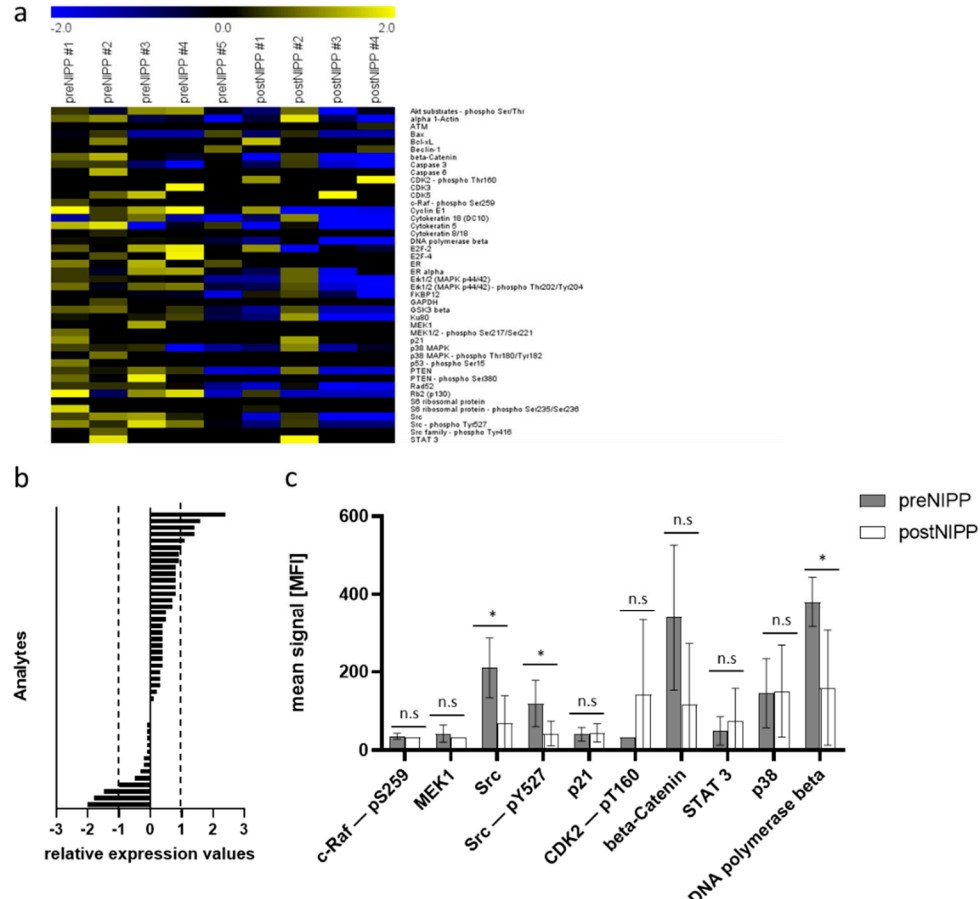

**Figure 4.** Protein expression after in vivo non-invasive physical plasma (NIPP) treatment of patients with cervical intraepithelial neoplasia (CIN). (**a**) Heat map of log2-transformed DigiWest data. Yellow indicates a high signal level and blue indicates a low signal level (compared to the median). (**b**) Bar graphs of log2-transformed ratios calculated from the mean expression of NIPP treatment and controls sorted from the highest positive change to the highest negative change. (**c**) Bar graphs of signals generated from patient samples. Shown are the means and standard derivation of NIPP treatment and controls. * indicates a significant difference in expression, n.s. indicates no significant difference in expression (Wilcoxon rank sum test, $p < 0.05$).

Interestingly, NIPP treatment of CSCC cells caused a significant decrease in the transcription factor STAT3, which plays an important role in many cellular processes, such as cell growth and apoptosis [24]. Furthermore, we found a downregulation of cyclin-dependent kinase inhibitor 1 (p21) being primarily associated with inhibition of cyclin-dependent kinase 2 (CDK2) [25,26]. Thus, the phosphorylation of CDK2 (CDK2-pT160) was also significantly increased, having an impact on cell growth and cell cycle regulation.

Tissue samples from in vivo NIPP-treated patients revealed comparable results. As shown in Figure 2, we were able to analyze 69 analytes using DigiWest technology. Forty-four of which were matching proteins of the previous analysis related to apoptosis machinery, DDR, and cell cycle control (Figures 3a and 4a). Overall, we detected higher differences in protein expression between when comparing FFPE tissue and cell lysates shown by more intensive signals in the heat map analysis of log2-transformed DigiWest data. Thereby, eight analytes in the FFPE sample set showed a considerable change in expression after in vivo NIPP treatment (mean log2 foldchange greater than 1 or −1) (Figure 4b).

Again, NIPP treatment of patients with CIN was accompanied by a significant decrease in proto-oncogene tyrosine-protein kinase Src expression and phosphorylation. Other pro-proliferative factors shown to be up- or downregulated (such as p38, c-Raf, MEK1, STAT3, and p21) were not significantly altered. Comparable to NIPP-treated CSCC cells, CDK2 showed a valuable increase; however, this was not statistically significant.

Additionally, protein profiling immediately after in vivo NIPP treatment (0 h) demonstrated a transient significant increase in cytokeratin 8 (CK8) and 18 (CK18) (Figure S2). After 24 h this effect was reversed. CK18 and the co-expressed complementary partner, CK8, maintain the physiological cell function against external stress and play an important role in apoptosis and the cell cycle [27]. Furthermore, we found a significant decrease in protein kinase B (Akt) expression 0 h and 24 h following in vivo NIPP treatment. The Akt signal transduction pathway promotes cell survival and cell growth in response to extracellular signals by regulating apoptosis and cell cycle [28,29]. Among other functions, Akt regulates the CDK inhibitor p21 and the proto-oncogene Src, promoting cell cycle progression.

## 4. Discussion

To date, only a few individual case reports exist, describing in vivo NIPP treatment of cancer patients [6,8,14]; none have been conducted yet on NIPP treatment of precancerous diseases. Recently, we established the electrosurgical argon plasma device VIO3/APC3 (Erbe Elektromedizin, Tübingen, Germany) for the in vivo treatment of patients [9,30]. Currently, we are performing a prospective, single-armed phase IIb clinical trial (NCT03218436) at the Department for Women's Health, Tübingen, Germany. For this purpose, the molecular examination of tiny cervical punch biopsies obtained from in vivo NIPP-treated patients is becoming more and more crucial. However, although fresh, frozen tissue would be ideal for research purposes, this is not practical for a clinical setting. For histopathological assessment, the morphological structure of tissues must be conserved. Therefore, clinical samples are immediately fixed with formalin and embedded in paraffin wax. Formalin cross-links form a protein grid that preserves the tissue structure and prevents protein degradation [31]. Subsequent paraffin embedding facilitates the handling of samples and sectioning of the tissue into thin slices for staining and microscopical assessment. Furthermore, formalin fixation enables long-term storage at room temperature. As efficient as formalin fixation is for the prevention of tissue degradation, the occurring protein crosslinks disturb most bioanalytical methods. In particular, the separation of molecules based on their molecular size is usually disabled, since the crosslinks cause irreversible protein aggregation, resulting in low amounts of specific proteins. However, DigiWest technology enables Western-blot-like incubation of up to app. 200 antibodies from minimal sample amounts by transferring proteins onto microspheres and miniaturization of the assay system [18].

In the present study, we established the targeted protein analysis from small-sized FFPE tissue sections obtained from cervical punch biopsies utilizing DigiWest technology. This enabled the analysis of molecular tissue effects following in vivo NIPP treatment of CIN. Moreover, we compared the results with the in vitro NIPP-treated human malignant cervical cell line, CSCC. Non-thermally operated NIPP devices lead to the formation of ROS, as shown by previous studies [6,14,32].

ROS and RNS are the responsible drivers of NIPP-related anti-neoplastic efficacy in human cervical cancer cells due to cell cycle arrest and apoptosis [9]. Additionally, in this study, NIPP treatment affected central factors in the regulation of apoptosis and cell growth pathways. In particular, NIPP treatment resulted in the transient induction of cell survival factors (CK8/18), accompanied and followed by the downregulation of pro-proliferative factors (here: Akt, p38 MAPK, Src, and RAF) and the upregulation of cell-growth-attenuating pathways. The in vitro cell panel used in this study includes cells from squamous epithelial tumors and adenocarcinomas. SiHa and CaSki cells harbor HPV infections; DoTc2 4510 cells originate from a metastatic CC lesion. Moreover, several well-known mutations of gynecological cancers, such as p53, BRCA2, or PIK3CA, are represented by these cell lines. In general, we found no evidence for a distinct factor resulting in increased NIPP resistance. This indicates a multifactorial intracellular process initiated by NIPP treatment.

Besides its structural function in the cytoskeleton, CK8/18 regulates apoptosis and is released during apoptosis and necrosis [33,34]. CK18 release can even occur independently of caspase activation [35]. Hence, our findings suggest that NIPP treatment may directly induce a cell survival response, followed by apoptosis.

It is likely that NIPP impairs the Akt-driven progression of the G1-S cell cycle phase by inactivating glycogen synthase kinase 3 (GSK-3) and preventing cyclin D1 degradation [36]. CDK2 is a subunit of the cyclin-dependent kinase complex, mainly involved and restricted to the regulation of the G1-S phase of the cell cycle [37]. The fully active CDK2 (in a complex with cyclins) is phosphorylated at threonine 160 (T160) [38], a regulative response which could also be shown after in vitro and in vivo NIPP treatment. Moreover, a downregulation of the CDK2 inhibitor p21 [25,26] was demonstrated in NIPP-treated CSCC cells. However, CDK2 does not seem to be essential for proceeding or arresting the transition during the G1-S phase [39].

Noticeable, in vitro and in vivo NIPP treatment caused a significantly reduced expression and phosphorylation of the proto-oncogene tyrosine-protein kinase Src, which was shown to be upregulated in about half of the tumors from the colon, liver, lung, breast, and pancreas [40]. Thereby, Src has a central impact on cell survival, proliferation, and invasion. Src has been shown to be involved in further pro-proliferative cell responses and to interact with important regulative factors including MAPK/MEK1/RAF, Akt, and STAT3 [22,23]. Interestingly, all of these interacting factors have been shown to be reduced after in vitro NIPP treatment of CSCC cells; however, it could be not confirmed after in vivo NIPP treatment of CIN. The results suggest that changes in protein expression observed in cell culture experiments may not be transferable to in vivo treatment.

This underlines the importance of performing studies on in vivo NIPP applications and consecutive analysis of the biological effects, some of which differ considerably from in vitro results. Overall, the analysis of cell culture samples is much easier; additionally, here, it resulted in more statistically significant changed analytes than the analysis of FFPE patient samples. However, the changes in expression after treatment were more distinct in the patient samples. This may be because FFPE samples delivered lower absolute intensities of signals compared to cell culture samples, which may be rooted in the differences in sample preparation. However, normalization to the total protein loaded on beads relativizes such effects. Additionally, the smaller samples size in this study may limit the statistical power of the used tests.

In conclusion, we demonstrated the molecular efficacy of NIPP treatment within human malignant cervical cell lines and CIN. NIPP effects were primarily based on the inhibitory impact on the cell cycle and cell growth factors. NIPP treatment effects need to be studied more frequently in vivo, or at least in patient-derived cell culture models such as organoids or patient-derived microtumors (PDMs) that mimic the in vivo situation much better [41]. DigiWest technology enables comprehensive protein profiling from very small and FFPE primary tissue biopsies.

**Supplementary Materials:** The following are available online at https://www.mdpi.com/article/10.3390/app112311238/s1, Figure S1: Overview of FFPE extraction and DigiWest workflow; Figure S2: Selected DigiWest results of a control (untreated) as well as 0 h and 24 h after treatment FFPE samples; Table S1: Raw and normalized data used for analysis; Table S2: List of used antibodies.

**Author Contributions:** Conceptualization, F.R., M.T. and M.W.; methodology, F.R., M.T. and M.W.; formal analysis, F.R. and M.W.; investigation, F.R. and M.W.; data curation, F.R. and M.W.; writing—original draft preparation, F.R. and M.W.; writing—review and editing, M.T., M.H., M.E., H.B., D.W., S.Y.B. and K.S.-L.; supervision, M.T., D.W., S.Y.B. and K.S.-L.; project administration, M.W. All authors have read and agreed to the published version of the manuscript.

**Funding:** This study was financially supported by the Faculty of Medicine of the Eberhard Karls University of Tübingen (grant No. 2432-1-0, 417-0-0 to M.W., and IZKF 2018-1-06 to M.W.), Graduate School 2543/1 "Intraoperative Multi-Sensory Tissue-Differentiation in Oncology" (project(s) A3 and C2) funded by the German Research Foundation (GRK 2543/1 to S.Y.B., K.S.-L., and M.W.; 04/2020).

We acknowledge support from the Open Access Publishing Fund of the University of Tübingen. This work received financial support from the State Ministry of Baden-Wuerttemberg for Economic Affairs, Labour and Tourism.

**Institutional Review Board Statement:** The study was conducted according to the guidelines of the Declaration of Helsinki, and approved by the Institutional Ethics Committee of the Medical Faculty of the Eberhard-Karls University of Tübingen (protocol code 237-2017BO1).

**Informed Consent Statement:** Informed consent was obtained from all subjects involved in the study.

**Data Availability Statement:** The data presented in this study are available in Table S1. Information about antibodies used in this study are available in Table S2.

**Acknowledgments:** This work was supported by Erbe Elektromedizin GmbH, Tübingen (loaner of NIPP device and equipment).

**Conflicts of Interest:** M.E. is an employee of Erbe Elektromedizin GmbH.

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
