# Peer review of "Targeted Protein Profiling of In Vivo NIPP-Treated Tissues Using DigiWest Technology"

_applsci, doi:10.3390/app112311238_

Round 1

Reviewer 1 Report

The authors present an interesting manuscript with a novel approach for the determination of molecular changes in tissue after treatment with non-thermal physical plasmas. By multiplex protein profiling, minimum samples sizes taken after in vivo treatments are sufficient to determine effects that to date were investigated only in vitro. The authors successfully demonstrate the analytical approach in comparison of in vivo, in vitro and literature results for changes brought about by the physical plasma treatment. Thereby, effects of reactive oxygen and nitrogen species reported in earlier studies were positively confirmed. Further, specific cell survival and pro-proliferation factors were identified that are impacted in the particular case of CIN lesions. Despite a comparatively limited number of patients and correspondingly low statistics, the presented results appear indicative and the conclusions justified.

Author Response

We thank reviewer #1 for his comments. 

Reviewer 2 Report

This paper shows the results of the molecular evaluation of the treatment of precancerous lesions of the cervix with non-invasive physical plasma (NIPP) using DigiWest technology, a high throughput bead-based Western-blot. The proposed technology allowed to assess the detailed protein profiles of FFPE small biopsies and demonstrated the molecular efficacy of NIPP treatment in human cervical malignant cell lines and in patients with CIN.

The study is interesting, the methodologies and results are described in adequate detail, but the authors should clarify the following points:

  1. Some of the titles in the section "Materials and Methods" do not correspond to what is described in the respective subsection: the subsection "2.1 In vivo NIPP treatment" in addition to in vivo also refers to in vitro treatment in cell culture plates. The same happens with the subsection "2.2 Propagation and in vitro NIPP treatment of cells" which only describes the propagation and not the treatment of cells in vitro.
  2. The authors report that they used CSCC acquired from ATCC, reference ATCC® TCP-1022. This ATCC reference refers to a panel of 4 cervical tumor cell lines with varying degrees of genetic complexity. The authors should clarify which cell line or lines they used as well as its molecular characterization. This characterization will be especially important for the interpretation of results.
  3. Still in relation to cell cultures, these cell lines are all adherent. However, in section 2.1, the authors refer to "Cells were dynamically treated in suspension on a 6-well cell culture plate...". The authors should clarify this statement.
  4. Some of the references do not appear to be complete. Please check in accordance with the journal rules.

Author Response

Reviewer 2; Comment 1

    Some of the titles in the section "Materials and Methods" do not correspond to what is described in the respective subsection: the subsection "2.1 In vivo NIPP treatment" in addition to in vivo also refers to in vitro treatment in cell culture plates. The same happens with the subsection "2.2 Propagation and in vitro NIPP treatment of cells" which only describes the propagation and not the treatment of cells in vitro.

  • We thank reviewer #2 for his comment. As recommended, we clarified the methodology of in vivo and in vitro NIPP treatment and rewrote the paragraphs 2.1 and 2.2.

Reviewer 2; Comment 2

    The authors report that they used CSCC acquired from ATCC, reference ATCC® TCP-1022. This ATCC reference refers to a panel of 4 cervical tumor cell lines with varying degrees of genetic complexity. The authors should clarify which cell line or lines they used as well as its molecular characterization. This characterization will be especially important for the interpretation of results.

  • We thank reviewer #2 for his comment. First, we defined the cell lines used in this study and added information about HPV infection into section 2.2:

[..]. In detail these were CaSki (ATCC CRL-1550), DoTc2-4510 (ATCC CRL-7920) and SiHa (ATCC HTB-35). CaSki and SiHa cells are positive for human papillomavirus (HPV) and are derived from squamous cell carcinomas of the cervix uteri, whereas DoTc2 4510 are derived from adenocarcinomas. [..]

  • Additionally, we discussed the possible influence of the different genetic mutation pattern of the used cell lines by adding the following paragraph into the discussion:

 [..]. The in vitro cell panel used in this study includes cells from squamous epithelial tumors and adenocarcinomas. SiHa and CaSki cells harbor HPV infections, DoTc2 4510 originate from a metastatic CC lesion. Moreover, several well-known mutations of gynecological cancers, such as p53, BRCA2, or PIK3CA are represented by these cell lines. In general, we found no evidence for a distinct factor resulting in increased NIPP resistance. This indicates a multifactorial intracellular process initiated by NIPP treatment. [..]

Reviewer 2; Comment 3

    Still in relation to cell cultures, these cell lines are all adherent. However, in section 2.1, the authors refer to "Cells were dynamically treated in suspension on a 6-well cell culture plate...". The authors should clarify this statement.

  • We thank reviewer #2 for his note. To further clarify the statement, we included the following paragraph into section 2.1:

 The NIPP treatment of single cells was performed in suspension for the following reasons: i) to avoid mechanical detachment and associated cell damage, as well as drying effects due to the NIPP gas flow and ii) to enable the NIPP treatment on a sterile grounded metal mold with identical electrical current and resistance. It was recently shown that plasma activation of media (plasma-activated media; PAM) reveals very similar anti-proliferative cell effects compared to direct plasma treatment [Koensgen et al. Anticancer Res. 2017, 37, 6739–6744]. Moreover, the unphysiological suspension state was limited to the treatment period before enabling the immediate reattachment of the cells.

Reviewer 2; Comment 4

    Some of the references do not appear to be complete. Please check in accordance with the journal rules.

  • We thank reviewer #2 for his thorough review of the manuscript and for identifying the incomplete reference information. We completely reviewed the citations for any missing information.
